# YAP/TEAD1 Complex Is a Default Repressor of Cardiac Toll-Like Receptor Genes

**DOI:** 10.3390/ijms22136649

**Published:** 2021-06-22

**Authors:** Yunan Gao, Yan Sun, Adife Gulhan Ercan-Sencicek, Justin S. King, Brynn N. Akerberg, Qing Ma, Maria I. Kontaridis, William T. Pu, Zhiqiang Lin

**Affiliations:** 1Masonic Medical Research Institute, 2150 Bleecker St, Utica, NY 13501, USA; gaoyunan2014@163.com (Y.G.); ysun@mmri.edu (Y.S.); gercansencicek@mmri.edu (A.G.E.-S.); mkontaridis@mmri.edu (M.I.K.); 2Department of Cardiology, The Fourth Affiliated Hospital of Harbin Medical University, Harbin 150001, China; 3Department of Neurosurgery, Program on Neurogenetics, Yale School of Medicine, Yale University, New Haven, CT 06510, USA; 4Department of Cardiology, Boston Children’s Hospital, 300 Longwood Ave, Boston, MA 02115, USA; kingjustinscott@gmail.com (J.S.K.); brynn.akerberg@childrens.harvard.edu (B.N.A.); Qing.Ma@childrens.harvard.edu (Q.M.); william.pu@enders.tch.harvard.edu (W.T.P.)

**Keywords:** YAP, TEAD1, Toll-like receptor, heart, TLR4, cardiomyocyte, innate immune responses

## Abstract

Toll-like receptors (TLRs) are a family of pattern recognition receptors (PRRs) that modulate innate immune responses and play essential roles in the pathogenesis of heart diseases. Although important, the molecular mechanisms controlling cardiac TLR genes expression have not been clearly addressed. This study examined the expression pattern of *Tlr1*, *Tlr2*, *Tlr3*, *Tlr4*, *Tlr5*, *Tlr6*, *Tlr7*, *Tlr8*, and *Tlr9* in normal and disease-stressed mouse hearts. Our results demonstrated that the expression levels of cardiac *Tlr3*, *Tlr7*, *Tlr8*, and *Tlr9* increased with age between neonatal and adult developmental stages, whereas the expression of *Tlr5* decreased with age. Furthermore, pathological stress increased the expression levels of *Tlr2*, *Tlr4*, *Tlr5*, *Tlr7*, *Tlr8*, and *Tlr9*. Hippo-YAP signaling is essential for heart development and homeostasis maintenance, and YAP/TEAD1 complex is the terminal effector of this pathway. Here we found that TEAD1 directly bound genomic regions adjacent to *Tlr1*, *Tlr2*, *Tlr3*, *Tlr4*, *Tlr5*, *Tlr6*, *Tlr7*, and *Tlr9*. In vitro, luciferase reporter data suggest that YAP/TEAD1 repression of *Tlr4* depends on a conserved TEAD1 binding motif near *Tlr4* transcription start site. In vivo, cardiomyocyte-specific YAP depletion increased the expression of most examined TLR genes, activated the synthesis of pro-inflammatory cytokines, and predisposed the heart to lipopolysaccharide stress. In conclusion, our data indicate that the expression of cardiac TLR genes is associated with age and activated by pathological stress and suggest that YAP/TEAD1 complex is a default repressor of cardiac TLR genes.

## 1. Introduction

Heart failure is one of the leading causes of mortality and morbidity in the developed countries [1]. Involved in the pathogenesis of heart failure, Toll-like receptors (TLR) are a family of pattern recognition receptors that sense pathogenic stimuli and signal the cardiac residential cells to cope with harsh conditions [2]. Human and mouse genomes contain 10 (*TLR1–10*) and 13 (*Tlr1–13*) TLR genes, respectively. *Tlr11*, *Tlr12*, and *Tlr13* exist in mouse but not in the human genome, and murine *Tlr10* is a pseudogene [3]. In the heart, TLR genes are expressed in both cardiomyocytes (CMs) and non-CMs [4], and disturbance of TLR genes expression has been implicated in a range of heart diseases, including pathogen and non-pathogen related heart failure [5,6]. Despite the importance of TLR genes in the pathogenesis of heart failure, the molecular mechanisms that regulate these genes’ expression are largely unknown.

Activation of CM innate immune signaling is one of the underlying mechanisms of ischemic and non-ischemic heart failure [7]. As crucial pattern recognition receptors, TLRs sense extracellular or intracellular danger signals and activate innate immune responses that lead to the synthesis and release of pro-inflammatory cytokine peptides [8]. The roles of several TLRs have been studied in the CMs, laying out a consensus direction that activation of TLRs is detrimental. For example, depletion of TLR4 attenuates myocardial infarction injury [9], whereas activation of TLR2, or TLR4, or TLR5 impairs CMs contractility [4]. Among the TLR pathways, TLR4/NF-κB signaling axis has been best established. TLR4 signals through a series of adaptor proteins and kinases to activate NF-κB, a central transcriptional driver of innate immune responses [10]. A canonical activator of TLR4 is bacterial pathogens-derived lipopolysaccharide (LPS), and LPS stress has frequently been used as the prototypical model to study innate immune responses in CMs [11] and other cell types [12].

Recently, cross talks have been found between TLR signaling and the Hippo-YAP pathway [13], an integral pathway regulating organ growth [14]. The already defined communications between these pathways happen through post-translational mechanisms, with YAP being one of the crucial node molecules. As the terminal effector of the Hippo-YAP pathway, YAP interacts with several transcription factors, such as TEAD1, to potently stimulate proliferation and promote cell survival [15]. We have recently reported that cardiomyocyte-specific YAP depletion up-regulated *Tlr2* and *Tlr4* [16]; however, it is unknown whether YAP directly regulates these two TLR genes through TEAD1 and whether YAP/TEAD1 complex also regulates other TLR genes. Addressing these questions will add a new layer of regulation between Hippo-YAP and TLR pathways and provide novel insights towards understanding the pathogenesis of heart failure.

This study documented the expression patterns of nine murine TLR genes (*Tlr1–9*) in developing mouse hearts and disease-stressed adult mouse hearts, studied the relationship between YAP/TEAD1 complex and TLR genes, and investigated YAP’s role in the regulation of cardiomyocyte innate immune signaling. Our results suggest that YAP/TEAD1 complex is a default repressor of cardiomyocyte TLR genes and indicate that YAP is required to restrain cardiomyocyte innate immune responses.

## 2. Results

### 2.1. The Expression of TLR Genes Increases with Age during Postnatal Heart Development

TLRs have been well-known as crucial players regulating cardiac inflammation [17], and recent data have also implicated TLR signaling in CM maturation [18]. Therefore, it is pivotal to delineate TLR gene expression patterns in the intact and disease-stressed heart. The TLRs have diverse cellular distributions and an array of ligands. TLR1, TLR2, TLR4, TLR5, and TLR6 are localized on the cell membrane and recognize bacterial and viral pathogen-associated molecular patterns in the extracellular matrix. TLR3, TLR7, TLR8, and TLR9 are intracellular TLRs associated with endosomes and recognize viral or bacterial genetic materials (Figure 1A) [19].

The molecular mechanisms regulating the expression of these TLR genes have not been sufficiently addressed. We analyzed whole-genome RNA sequencing data from fetal and adult murine hearts [20] for expression of the TLR genes (Figure 1B). The cardiac expression of *Tlr5* was higher in the fetal than in the adult heart, and the expression of *Tlr1*, *Tlr6*, *Tlr9*, *Tlr12* did not differ significantly between these two developmental stages. The remaining TLR genes all had lower expression levels in the fetal heart (Figure 1B). To further analyze the expression dynamics of these TLR genes during postnatal heart development, we selected four time points: embryonic day 18.5 (E18.5), postnatal day 5 (P5; Neonate), P14 (Juvenile), and P42 (Adult). Among the five genes encoding cell membrane TLRs, *Tlr2* and *Tlr4* increased with age, *Tlr5* decreased with age, and *Tlr1/Tlr6* fluctuated among the four developmental stages (Figure 1C). Different from the cell membrane TLRs, the expression of intracellular TLR genes, including *Tlr3*, *Tlr7*, *Tlr8*, and *Tlr9*, showed an increasing trend during heart development (Figure 1D). Here we summarized the expression level order of the four intracellular cardiac TLR genes: *Tlr3* > *Tlr8*, *Tlr9* > *Tlr7* (Figure 1D).

### 2.2. The Expression of TLR Genes in Disease-Stressed Hearts

To test whether non-pathogen stress affects TLR gene expression, we examined the mRNA levels of *Tlr1–9* in pressure overload (PO) and ischemia/reperfusion (IR) stressed mouse hearts. PO stress was induced by transverse aortic constriction (TAC) surgery, and hearts were collected ten days after TAC. IR stress was triggered by occluding the left anterior descending (LAD) coronary artery for 50 min, and hearts were collected two days after IR surgery. Compared with TLR genes in sham controls, *Tlr2*, *4*, *7*, *8*, and *9* were significantly increased in the PO-stressed hearts (Figure 2A). In IR-stressed hearts, *Tlr2*, *4*, *5*, and *9* were significantly increased (Figure 2B). These data suggest that TLR genes are largely primed in disease-stressed hearts.

Because TLR4 is the best studied TLR and validated commercial TLR4 antibody is available, we corroborated our observation by examining the protein level of TLR4. In sham control myocardium, TLR4 protein was only detected in non-CMs (Figure 2C). However, in either IR- or PO-stressed myocardium, cardiomyocyte TLR4 was readily detectable (Figure 2D). Taken together, these data suggest that TLR signaling pathways may be hyperactive in PO- or IR-stressed hearts.

### 2.3. Cardiomyocyte Specific YAP Depletion Does Not Induce Cardiac Hypertrophic Remodeling in the First 12 Days after Birth

Because TLRs play essential roles in regulating cardiac inflammation [8], it is pivotal to understand the molecular mechanisms that govern their expression. We previously reported that cardiomyocyte-specific YAP depletion increased the expression of *Tlr2* and *Tlr4* [16] and that cardiac YAP/TEAD1 complex decreased with age [21]. We then hypothesized that YAP/TEAD1 restrained the expression of cardiac TLR genes. To test this hypothesis, we generated cardiomyocyte-specific *Yap* knockout (*Yap^cKO^*) mice by crossing *Myh6::Cre* transgenic mice [22] to *Yap* flox allele [23] (Figure 3A). Cardiomyocyte specific YAP depletion causes dilated cardiomyopathy in adult mice [24], and the associated hypertrophic remodeling process may confound TLR gene expression analysis. To avoid this potential culprit, we determined to find a developmental stage at which the hypertrophic remodeling process had not started in the *Yap^cKO^* heart.

Compared to littermate control, *Yap^cKO^* mice had a lower heart-to-body weight ratio trend at postnatal day 7 (P7), which reached significance at P12. Because YAP/TEAD complex is required for CM proliferation [25,26], and some CMs are still proliferative in the neonatal heart [27], it is possible that the lower heart-to-body weight ratio of P12 *Yap^cKO^* mice is due to less CM proliferation. At P35, the heart-to-body weight ratio between control and *Yap^cKO^* mice was not distinguishable (Figure 3B). These heart-to-body weight ratio data indicate that compensatory cardiac hypertrophy happens between P13 and P35. In support of this notion, no increase in cardiac fibrosis was observed in the *Yap^cKO^* heart at P12 (Figure 3C). At this development stage, YAP knockdown was confirmed by western blot (Figure 3D), and the presence of residual YAP might be contributed by non-CMs [21] or due to the young age of *Yap^cKO^* mice. We then checked YAP expression in 2-month-old hearts and found that YAP was efficiently knocked down at this age (Appendix A).

Interestingly, TEAD1 protein was decreased in *Yap^cKO^* hearts (Figure 3E), suggesting that loss of YAP destabilizes TEAD1. Together, these data suggest that knocking out YAP does not induce cardiac hypertrophic remodeling in the first 12 days after birth. Therefore, we used P12 hearts for TLR gene expression analysis in the following studies.

### 2.4. YAP/TEAD1 Complex Regulates the Expression of TLR Genes

To determine whether YAP/TEAD1 complex directly regulates the expression of TLR genes, we analyzed recently reported fetal and adult cardiac TEAD1 chromatin immunoprecipitation and high throughput sequencing (ChIP-seq) data for TEAD1 binding to regions neighboring TLR genes [20]. In the fetal heart, TEAD1 directly bound to the promoters of *Tlr2*, *Tlr4*, *Tlr5*, *Tlr6*, *Tlr7*, and to regions neighboring *Tlr1*, *Tlr2*, *Tlr5*, *Tlr9* (Figure 4A). In the adult heart, TEAD1 bound to the promoters of *Tlr3*, *Tlr4*, *Tlr6*, and to regions neighboring *Tlr1*, *Tlr5*, and *Tlr9* (Figure 4A). For *Tlr2* and *Tlr7*, no TEAD1 ChIP seq peaks were detected in adult hearts. In addition, in both fetal and adult hearts, TEAD1 ChIP seq occupancy was not detected adjacent to *Tlr8* (Appendix A).

We then measured the expression of *Tlr1–9* with qRT-PCR. Our results showed that except for *Tlr6*, all the other eight TLR genes were significantly up-regulated in *Yap^cKO^* hearts (Figure 4B, Appendix A). Together, these data suggest that YAP/TEAD1 complex is a master suppressor of cardiac TLR genes.

### 2.5. YAP/TEAD1 Complex Directly Suppresses the Expression of Tlr4

The TEAD1 ChiP-seq and *Yap^cKO^* gene expression data suggest that YAP/TEAD1 complex directly regulates the expression of TLR genes. To validate this hypothesis, we used *Tlr4* as a prototype gene for further analysis. The TEAD1 ChIP-seq peak of *Tlr4* spans its transcription start region (TSSR), which contains a conserved TEAD1 binding motif 104 bp downstream of the transcription start site (Figure 5A). We first did ChIP-qPCR to validate TEAD1 binding of this region. The TEAD1 ChIP products of adult hearts were amplified by two sets of primers: the first set of primers spanning the *Tlr4* TSSR and the second set of primers annealing to the region 2.5 kb upstream of *Tlr4* TSSR. qPCR amplicons from the first but not from the second primer set were enriched in the ChIP products (Figure 5B), confirming the direct binding of TEAD1 to *Tlr4* TSSR.

Next, we cloned the human TLR4 TSSR and its adjacent region into a luciferase reporter vector, TLR4_Luci. We created another reporter, TLR4_Mut_Luci, by mutating the TEAD1 binding motif sequence AGAATGC into AtccaaC (Figure 5C). If YAP/TEAD complex was required for suppressing TLR4 expression, disrupting YAP/TEAD interaction would activate TLR4_Luci but not TLR4_Mut_Luci reporter. In 293 T cells, TLR4_Mut_Luci reporter had significantly higher activity than TLR4_Luci (Figure 5D), indicating that the conserved TEAD1 binding motif is required for suppressing TLR4 expression. Consistently, disrupting YAP and TEAD interaction with a YAP-TEAD interfering peptide (YTIP) [26] significantly increased the activity of TLR4_Luci but had no effects on TLR4_Mut_Luci (Figure 5E), indicating that YAP/TEAD complex is required to repress TLR4 expression.

### 2.6. CM-Specific YAP Depletion Activates TLR4/NF-κB Signaling

In the heart, loss of either YAP or TEAD1 decreased CM survival and caused heart failure, probably due to the downregulation of anti-apoptosis genes and disruption of mitochondria structure [24,28]. Here, our data indicate another mechanism that the TLR genes are primed in the YAP or TEAD1 knockout hearts. This sensitizes the CMs to environmental stresses, such as extracellular damage-associated molecular patterns (DAMPs) released by damaged cells and intracellular mitochondrial DNA escaped from impaired mitochondria. Because TLR4 is one of the best-characterized DAMPs receptors, and TLR4/NF-κB signaling axis plays essential roles in the pathophysiology of heart failure [10], we tested our hypothesis by focusing on analyzing whether loss of YAP resulted in activation of TLR4/NF-κB pathway.

Consistent with our expectation, TLR4 and RelA protein levels were both increased in P12 *Yap^cKO^* hearts (Figure 6A,B). We further measured the expression of four NF-κB target genes: *Ccl2* [29], *Il1b* [30], *Il12a* [31], and *Il12b* [32]. Except for *Ccl2*, the other three NF-κB target genes were significantly elevated in the *Yap^cKO^* hearts (Figure 6C). In situ immunohistochemistry staining confirmed that the synthesis of both IL1β and IL12β was increased in the *Yap^cKO^* myocardium (Figure 6D). Together, these data strongly suggest that cardiomyocyte-specific YAP depletion activates TLR4/NF-κB signaling pathway.

### 2.7. Knocking Down YAP in the CMs Predisposes the Heart to Lipopolysaccharide (LPS) Stress

Our current data suggest that YAP is required to blunt CMs innate immune signaling. We tested whether knocking down YAP predisposed the heart to acute LPS stress to further validate this hypothesis. Because whole-heart YAP depletion caused heart failure [24], we used an Adeno-associated virus (AAV) system to knock out YAP in a portion of CMs. AAV9.cTnT.iCre (AAV.iCre) [33] was retro-orbitally delivered to three weeks old *Yap^fl/fl^* mice at a dose of 5 × 10^9^ virus genomes (vg)/gram (g) body weight, which is sufficient to transduce 50–60% of the CMs [34]. *Yap^fl/fl^* mice receiving the same dose of AAV9.cTnT.GFP (AAV.GFP) were used as a control. At 21 days after AAV delivery, mice were treated with a sub-lethal dose of LPS (6 mg/kg) for 6 h [35]. Echocardiography measurements were performed before and after LPS treatment (Figure 7A).

Three weeks after AAV delivery, AAV.iCre + *Yap^fl/fl^* mice had similar fraction shortening with AAV.GFP + *Yap^fl/fl^* mice (Figure 7B,C). After LPS treatment, both AAV.iCre + *Yap^fl/fl^* and AAV.GFP + *Yap^fl/fl^* mice had reduced systolic heart function (Figure 7B,C). Compared to AAV.GFP + *Yap^fl/fl^* mice, AAV.iCre + *Yap^fl/fl^* mice were more vulnerable to LPS stress, as evidenced by significantly lower fraction shortening (Figure 7B,C). Nevertheless, the heart rate was not distinguishable between these two groups of mice before and after LPS treatment (Appendix A). At the molecular level, we confirmed that YAP was knocked down, and TLR4 and RelA were up-regulated in the AAV.iCre + *Yap^fl/fl^* hearts (Figure 7D, Appendix A). *Cyr61* is a well-defined YAP target [36]. Consistent with YAP knockdown, *Cyr61* was significantly reduced in AAV.iCre + *Yap^fl/fl^* hearts (Figure 7E). Additionally, *Tlr4* was significantly up-regulated in AAV.iCre + *Yap^fl/fl^* hearts (Figure 7E).

Activation of YAP has been reported to reduce LPS-induced CM apoptosis [37]. We then examined whether knocking down YAP increased LPS-induced CM apoptosis. Unlike the published study [37], we treated mice with LPS for 6 h instead of two days. In this condition, we did not detect CM apoptosis in both AAV.GFP + *Yap^fl/fl^* and AAV.iCre + *Yap^fl/fl^* hearts (Appendix A). Additionally, 6 h LPS treatment did not result in macrophages and neutrophils infiltration in either AAV.GFP + *Yap^fl/fl^* or AAV.iCre + *Yap^fl/fl^* hearts (Appendix A). These observations align with previous reports that acute LPS-TLR4 signaling decreases heart function by impairing CM contractility but not by inducing either CM loss [38,39] or leukocytes infiltration [9]. Additionally, together with Figure 6, our data suggest that knocking down YAP in CMs increases TLR4 expression, which predisposes the heart to LPS stress by sensitizing CMs to LPS-TLR4 signaling.

## 3. Discussion

This study investigated the expression patterns of nine cardiac TLR genes and studied the transcriptional relationship between YAP/TEAD1 complex and these genes. We found that the expression levels of most cardiac TLR genes were associated with age and activated by pathological stress. Furthermore, our data suggest that YAP/TEAD1 complex directly suppresses the expression of most cardiac TLR genes, and that loss of YAP primes CM innate immune gene expression programs by activating TLR4/NF-κB signaling (Figure 7F).

### 3.1. Expression of Cardiac TLR Genes Is Associated with Age and Activated by Pathological Stress

During postnatal heart development, CMs experience a maturation process [40], and non-CMs gradually expand their population [41]. Here, we found that the expression levels of most cardiac TLR genes were low in neonatal mice but significantly higher in adults. Three possible mechanisms underlie these observations: first, the expression of cardiac TLR genes increases due to CM maturation; second, the increase of cardiac TLR genes is due to the expansion of non-CMs; third, both CM maturation and non-CM expansion contribute to the upregulation of cardiac TLR genes. Because the percentage of non-CMs does not increase between E18.5 and postnatal day 14 [41], the upregulation of cardiac TLR genes should be mainly due to CM maturation during this growth period. Interestingly, the expression of all the intracellular TLR genes increases during this development phase, suggesting that upregulation of intracellular TLR genes is positively correlated with CM maturation.

In the adult heart, we examined the expression of TLR genes in two different heart disease models: PO-induced chronic cardiac hypertrophy and IR-triggered acute cardiac injury. Our data demonstrate that the expression of many cardiac TLR genes was activated by pathological stress. In this study, we used RNA from the whole heart to do gene expression analysis. In a more detailed study, after myocardial infarction (MI), cardiac tissues from the infarct zone and the remote zone were separated for TLR gene expression analysis. The results showed that TLR genes were increased in the infarct zone but kept unchanged in the remote zone [6], suggesting that upregulation of cardiac TLR genes may be mainly due to the expansion of non-CMs in disease-stressed adult hearts.

### 3.2. YAP/TEAD1 Complex Is a Default Repressor of Cardiomyocyte TLR Genes

Recently, we and others have shown that activating YAP in CMs suppressed the expression of inflammatory genes [16,42]. This study extended our understanding of YAP’s role in cardiac homeostasis by systemically interrogating the relationship between YAP and cardiac TLR genes. Our data demonstrated that TEAD1 is directly bound to the regions neighboring *TLR genes*, and CM-specific YAP depletion increased TLR gene expression. It is therefore conceivable that YAP/TEAD1 complex serves as a default suppressor of cardiomyocyte TLR genes. Of note, YAP might be required but not sufficient for suppressing TLR genes. For instance, YAP activation did not suppress *Tlr4* expression in either LPS-treated CMs or IR-stressed hearts [16]. Furthermore, some TLR genes were increased in IR-stressed hearts (Figure 2B), whereas nuclear YAP was enriched in the CMs of diseased hearts [24,43]. Therefore, YAP activation is likely insufficient to suppress TLR genes, and pathological stress may activate additional molecular mechanisms that overcome YAP/TEAD-mediated default repression of TLR genes.

Despite that activating YAP represses a subset of target genes by recruiting nucleosome remodeling and histone deacetylase (NuRD) complex [44] in MCF10A mammary epithelial cells, our current data suggest that this is unlikely the case for YAP/TEAD1 regulation of *Tlr4*. Interestingly, TEAD1 protein was decreased in *Yap^cKO^* hearts (Figure 3), suggesting that loss of YAP increases *Tlr4* expression through its effect on TEAD1 expression. Future studies are secured to elucidate the mechanism of how YAP/TEAD1 complex regulates TLR genes.

### 3.3. YAP Is Required for Blunting CM Innate Immune Signaling

Ischemic or non-ischemic pathological stress activates CM innate immune signaling pathways [45], which stimulate pro-inflammatory cytokine release and reactive oxygen species (ROS) production [7,46]. These innate immune responses are beneficial for defending CMs against pathogen invasion and tissue repair, but also cause cardiac inflammation and myocardial damage, including CM dysfunction or death. Our published [16] and current data have identified YAP as a crucial suppressor of TLR4-mediated innate immune responses in CMs. Consistent with published data [9], we showed that acute LPS stress reduced heart function before the onset of cardiac inflammation. We further showed that knocking down YAP in the heart activated TLR4/NF-κB signaling (Figure 6) and exacerbated LPS-induced cardiac shock (Figure 7). These data suggest that YAP in the CMs cell-autonomously protects the heart against LPS stress by blunting TLR4/NF-κB pathway.

Although we found that loss of YAP increased TLR4 and RelA expression, we did not reveal the underlying mechanism of how YAP suppresses TLR4/NF-κB signaling. The TLR4/NF-κB signaling axis comprises multiple adaptor proteins and kinases, such as MyD88, TRAF6, and TAK1 [10]. Recently, in a non-cardiac context, YAP has been shown to inhibit intracellular innate immune signaling independent of its transcriptional activity. Instead, YAP directly impedes the activity of crucial innate immune signaling components, such as IRF3 [47], TAK1 [48], TBK1 [49], and TRAF6 [50]. More work needs to be done to dissect YAP’s role in CM innate immune signaling, including a determination of whether YAP regulates CM innate immune responses through its transcriptional or non-transcriptional activity, or both. 

## 4. Material and Methods

Supplemental information provides expanded material and experimental procedures.

### 4.1. Experimental Animals

All animal procedures were approved by the Institute Animal Care and Use Committees of Masonic Medical Research Institute and Boston Children’s Hospital. All experiments were performed in accordance with NIH guidelines and regulations. C57BL/6J mice aged 6–8 weeks were obtained from Jackson Labs. Swiss Webster (CFW) mice aged 6–8 weeks were obtained from Charles River Laboratories. *Myh6::Cre* [22] and *Yap* flox alleles [23] were reported previously.

### 4.2. LPS Treatment

*E. coli* O55:B5 LPS (Sigma, St. Louis, MO, USA, #L2880) was dissolved in saline and sterile-filtered. LPS was intraperitoneally delivered at a dose of 6 mg/kg body weight. Then, 6 h after LPS delivery, mice were tested for cardiac function. The mice were sacrificed for tissue harvesting after echocardiography measurements.

### 4.3. Gene Expression

Total RNA was isolated using Trizol. For qRT-PCR, RNA was reverse transcribed (Applied Biological Materials Inc., Richmond, BC, Canada, G454), and specific transcripts were measured using Sybr Green chemistry (Lifesct., Rockville, MD, USA, LS01131905Y) and normalized to *Gapdh*. Primer sequences were provided in Appendix A. Primary antibodies used for immunoblot and immunohistochemistry staining were listed in Appendix A.

### 4.4. Statistics

Values were expressed as mean ± SD. In addition, student’s *t*-test or ANOVA with Tukey’s honestly significant difference post hoc test was used to test for statistical significance involving two or more than two groups, respectively.

## Figures and Tables

**Figure 1 ijms-22-06649-f001:**
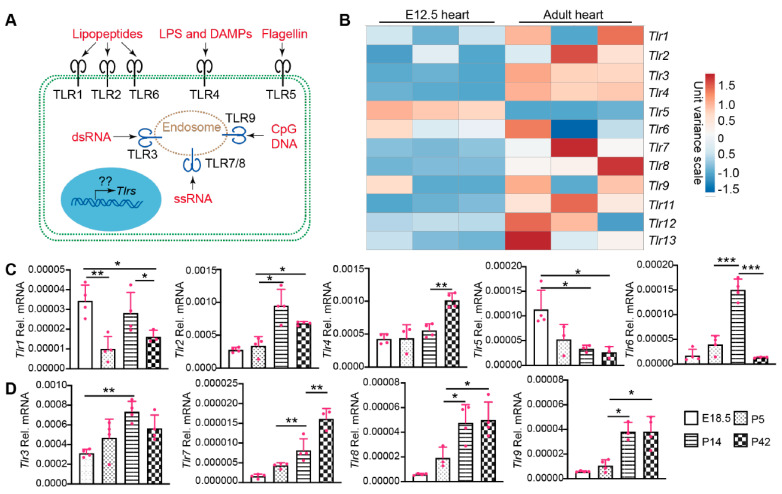
The expression of TLR genes in different age hearts. (**A**). Schematic view of TLR cellular localization and ligands. TLR1, 2, and 6 recognize bacteria- or mycoplasma-derived lipopeptides. TLR4 recognizes bacteria-derived lipopolysaccharide (LPS) and host-cell-derived death associated molecular patterns (DAMPs). Bacteria-derived Flagellin is the natural ligand of TLR5. Virus-derived double strand RNA (dsRNA) and single strand RNA (ssRNA) binds to and activates TLR3 and TLR7/8, respectively. TLR9 recognizes DNA containing unmethylated CpG motifs prevalent in microorganisms and mitochondrial DNA, but not in vertebrate genomic DNA. On the transcriptional level, mechanisms regulating the expression of these TLR genes are largely unknown. (**B**). Heat map of TLR gene expression levels in fetal and adult hearts. TLR gene expression values were retrieved from published RNA seq data (22). The heat map was generated with an online tool (ClustVis). (**C**,**D**). qRT-PCR measurement of TLR genes during heart development. Total RNA from hearts with the indicated ages was used for qRT-PCR. Gene expression level was normalized to *Gapdh*. E18.5, embryonic day 18.5; P5, P14, P42, postnatal days 5, 14, and 42. One-way ANOVA post hoc Tukey’s multiple comparisons test, *, *p* < 0.05; **, *p* < 0.01, ***, *p* < 0.001. *n* = 4.

**Figure 2 ijms-22-06649-f002:**
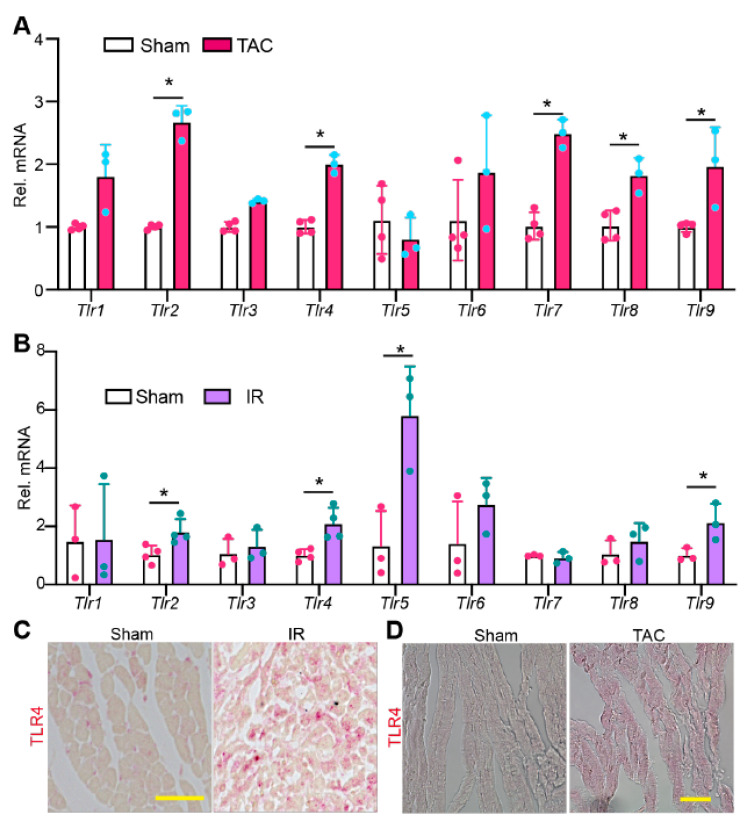
The expression pattern of TLR genes in stressed hearts. qRT-PCR measurement of TLR genes in pressure overload stressed hearts. 6–8 weeks old mice were stressed with TAC (transverse aortic constriction) surgery for ten days before being sacrificed for cardiac gene expression analysis. (**B**). qRT-PCR measurement of TLR genes in ischemia-reperfusion (IR) stressed hearts. Gene expression level was normalized to *Gapdh*. (**A**,**B**), student *t*-test, *, *p* < 0.05. *n* = 3–4. (**C**,**D**). Representative images of myocardium immunostained for TLR4. Scale bar = 50 µm. (**B**,**C**), hearts were collected for analysis two days after IR surgery.

**Figure 3 ijms-22-06649-f003:**
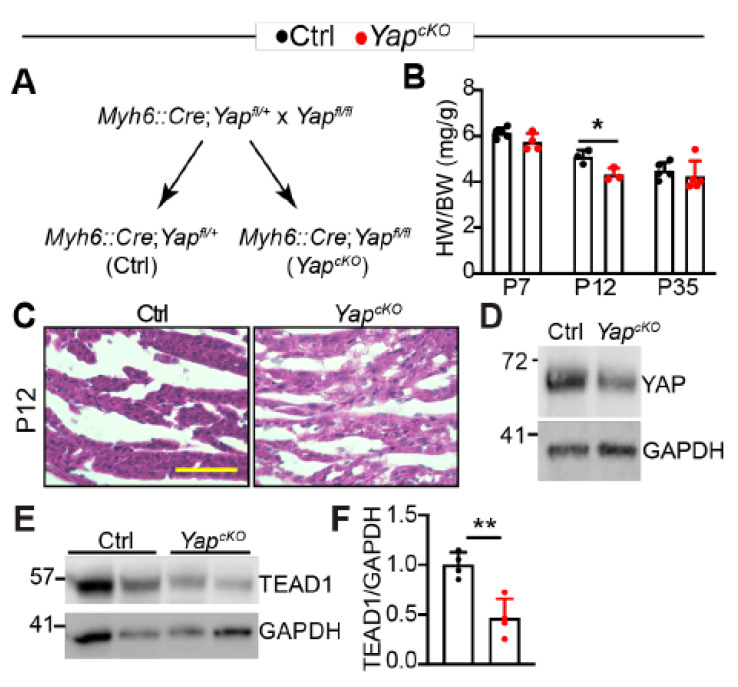
Cardiomyocyte-specific YAP depletion does not cause cardiac hypertrophy in mice less than 12 days of age. (**A**). Genetic strategy to knock out YAP in CMs. *Myh6::Cre; Yap^fl/+^* mice were used as control (Ctrl) in the subsequent studies. (**B**). Heart-to-bodyweight ratio. Heart and body weight was measured at postnatal day 7, 12, and 35. *n* = 3–5 for each group. (**C**). H&E staining of P12 control and *Yap^cKO^* hearts. Scale bar = 50 µm. (**D**). YAP immunoblot. (**E**). TEAD1 immunoblot. (**D**,**E**), total heart protein from P12 mice was used for immunoblot. **F**. Densitometry quantification of TEAD1. TEAD1 protein level was normalized to GAPDH. *n* = 3. (**B**,**F**), student’s t test, *, *p* < 0.05; **, *p* < 0.01.

**Figure 4 ijms-22-06649-f004:**
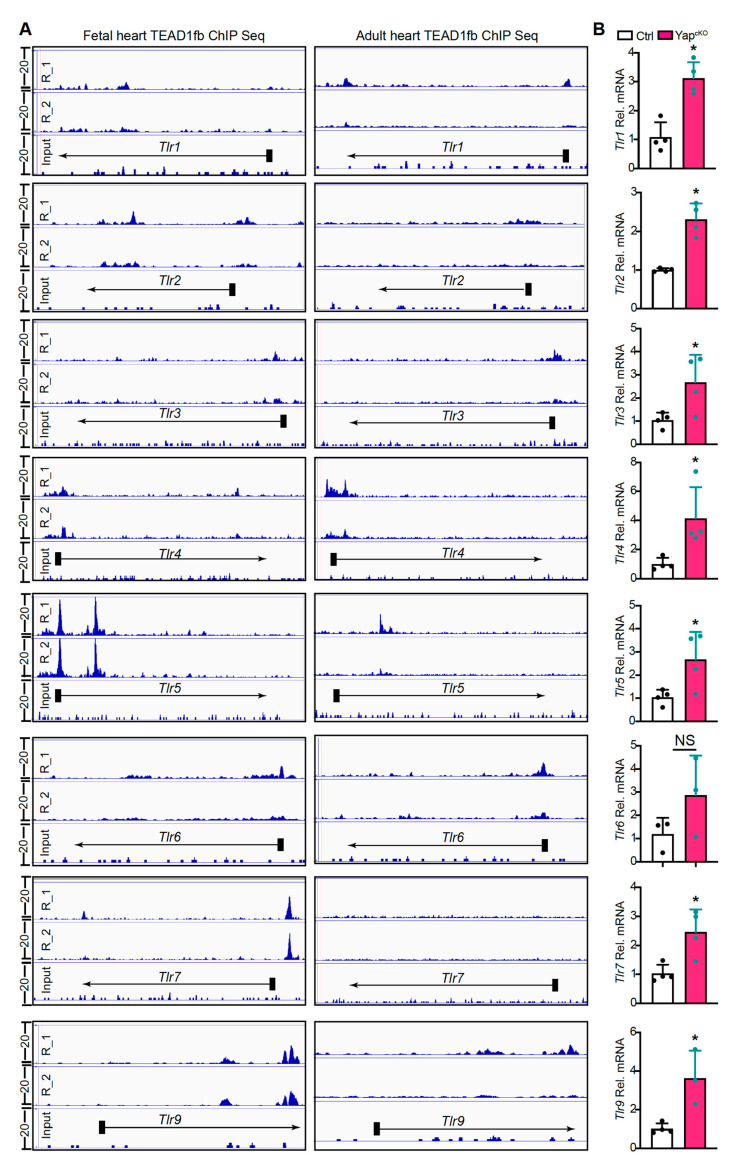
Cardiomyocyte-specific YAP depletion activates TLR genes expression. (**A**). Genome browser view showing chromatin immunoprecipitation and high throughput sequencing (ChIP-seq) of TEAD1 occupancy near the transcriptional start site (TSS) or gene body of the TLR genes. E12.5 fetal heart and adult hearts were used for TEAD1^fb^ ChIP-seq. ChIP-seq data from two biological replicates are sown. (**B**). qRT-PCR measurement of cardiac TLR genes. Gene expression level was normalized to *Gapdh*. *n* = 4. Student *t*-test, *, *p* < 0.05.

**Figure 5 ijms-22-06649-f005:**
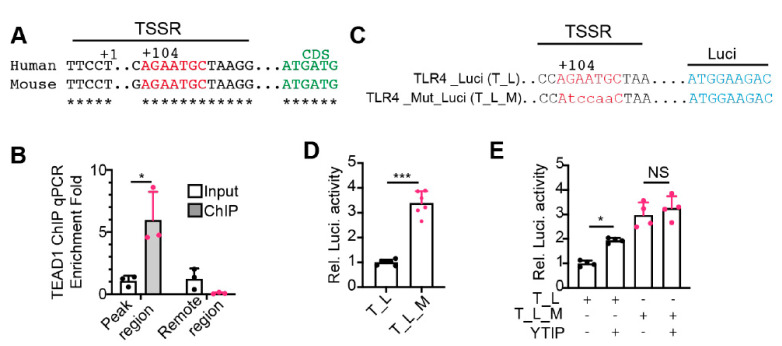
YAP/TEAD1 complex directly suppresses *Tlr4* expression. (**A**). Schematic view of the transcription start site (TSS, +1) regions of human and mouse TLR4 genes. The conserved TEAD1 binding motif was depicted with red letters, and the TLR4 coding sequences were indicated with green letters. (**B**). TEAD1 ChIP qPCR. A primer set spanning the mouse *Tlr4* TSS region was used to amplify ChIP products. Another primer set amplifying one fragment 2.5 kb away from the TSS was used as a negative control. Amplicon values from TEAD1 ChIP products were normalized with that of the input. *n* = 3. Student *t*-test, *, *p* < 0.05. (**C**). Depiction of the luciferase reporter constructs. TLR4_Luci (T_L) reporter contains the human TLR4 promoter driving luciferase. A conserved TEAD1 binding motif in the TLR4 promoter was mutated to generate TLR4_Mut_Luci (T_L_M). (**D**,**E**). Dual-luciferase reporter assay in HEK293T cells. Dual-luciferase assay was performed 24 h after transfection. YTIP: YAP and TEAD1 interference peptide. (**D**), student *t*-test, *, *p* < 0.05. (**E**), One-way ANOVA post hoc Tukey’s multiple comparisons test, *, *p* < 0.05, ***, *p* < 0.001. (**D**,**E**), *n* = 4–6.

**Figure 6 ijms-22-06649-f006:**
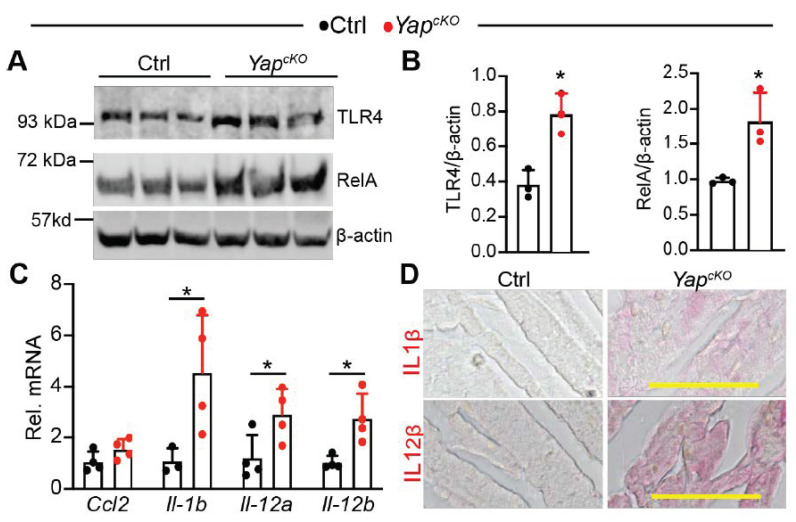
Cardiomyocyte-specific YAP depletion activates TLR4/NF-κB pathway. (**A**). Immunoblot of TLR4 and RelA. Total heart protein from postnatal day 12 (P12) mice was used for immunoblot. (**B**). Densitometry quantification of TLR4 and RelA. The protein levels of TLR4 and RelA were normalized to β-actin. *n* = 3 for each group. (**C**). qRT-PCR measurement of pro-inflammatory genes. Total RNA from P12 heart was used. Gene expression level was normalized to *Gapdh*. *n* = 4. (**B**,**C**), student *t*-test, *, *p* < 0.05. (**D**). Immunohistochemistry staining of P12 myocardium. Alkaline phosphatase-based detection system was used to visualize the expression of IL-1β and IL-12β. Scale bar = 50 µm.

**Figure 7 ijms-22-06649-f007:**
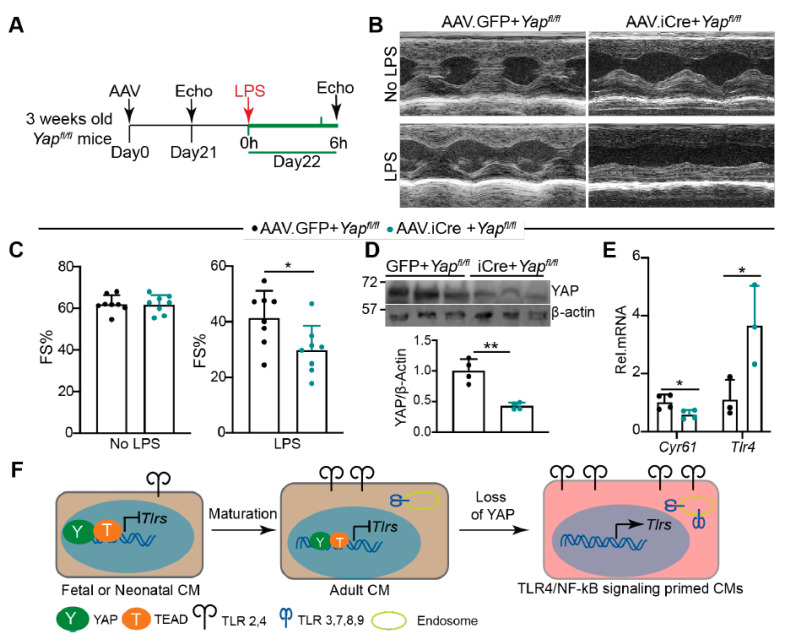
Knocking down YAP in CMs predisposes the heart to LPS stress. (A). Experimental design. Indicated AAVs were retro-orbitally delivered into 3 weeks old *Yap^fl/fl^* mice. Twenty-one days later, echocardiography was performed to measure cardiac function. Twenty-two days after AAV delivery, 6 mg/kg LPS was intraperitoneally injected into AAV transduced mice. Echocardiography measurements were performed 6 h after LPS treatment. (**B**). Representative M mode cardiac echocardiograms before and after LPS treatment. (**C**). Fraction shortening before and after LPS treatment. AAV9.cTnT.GFP (AAV.GFP), *n* = 8; AAV9.cTnT.iCre (AAV.iCre), *n* = 8. (**D**). YAP Immunot Blot and densitometry quantification. YAP protein level was normalized to β-actin. *n* = 4 for each group. (**E**). qRT-PCR measurement. *N* = 3–4 for each group. Gene expression level was normalized to *Gapdh*. (**D**,**E**), 6 h after LPS treatment, hearts were collected for protein and RNA extraction. (**C**–**E**), student’s t test, *. *p* < 0.05, **, *p* < 0.01. (**F**). Schematic summary of the current study. During heart development, the expression of *Tlr2* and *Tlr4* and intracellular TLR genes increases with age. YAP/TEAD1 complex is a default repressor of TLR genes in intact hearts. Cardiomyocyte-specific YAP depletion unleashes the expression of TLR genes and primes the CMs for generating inflammatory cytokine peptides. Brown color background in fetal/neonatal and adult CMs indicates a healthy status; pink color background in YAP depleted CMs indicates an innate immune signaling active status.

## Data Availability

The RNA-seq and ChIP-seq data presented in this study are openly available at NCBI GEO database (GSE124008), Reference doi: 10.1038/s41467-019-12812-3.

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
