# Peer review of "YAP/TEAD1 Complex Is a Default Repressor of Cardiac Toll-Like Receptor Genes"

_ijms, 2021, doi:10.3390/ijms22136649_

Round 1
Reviewer 1 Report
Gao and collaborators describe in the present manuscript the involvement of YAP/TEAD1 complex in TLR gene expression in heart. They have done a big effort to demonstrate it, presenting an excellent methodology, directed to what they want to explore. However, some minor concerns are present:
-Maybe I missed some information present in the manuscript, but it seems confusing that authors emphasise the control of YAP/TEAD1 over TLR gene expression in CM when their experiments are performed in total heart and at least TLR4 protein is not found in healthy adult heart . After heart disease, expression is found in both CM and non-CM fractions. Have authors separate CM and non-CM and analysed TLR gene expression?
- Is there any change related to TLRs expression from E12.5 to E18.5? Do not really understand the reason of performing the whole genome analysis (fig 1B) and CHIP (fig 4A) at E12.5and then change to E18.5 as the initial point for the qPCR assay (fig 1C-D).
- Please, specify the method used to analyse qPCR data in fig 1C-D. It seemed authors used different methods in this figure and fig 4B
- What happens with YAP/TEAD1 during development? Is there any correlation between its expression and TLR gene expression pattern from embryo to adult stage?
Reviewer 2 Report
In this manuscript, Gao and colleagues identify that the expression of nine murine Toll-like receptors (TLRs) in normal and stressed/perfused heart tissues. They report the increase of several TLRs that increased with cardiac perfusion and suggested that loss of YAP and its TF TEADs may provide a mechanistic basis for the transcriptional upregulation of TLRs. Overall, the manuscript is well written with results largely supporting their major claims. Furthermore, the apt usage of cardiac-specific YAP fl/fl mouse models and in-vivo demonstrations of YAP depletion using AAV that makes the adult heart tissues susceptible to increased stress is the major strength of this work. However, a few issues need to be addressed before progressing towards publication.
Specific comments:
1) In Fig 1b, a demonstration of respective mRNA expression of TEAD AND YAP/TAZ targets are missing? These should negatively correlate with TLRs expression.
2) What is the status of nuclear vs cytosolic YAP levels in perfused heart tissues in Fig 2?
3) Why do the cKO heart tissues (Fig 3) still express 50% YAP? is the expression system leaky? This should be explained in the text.
4) Fig 7d: the protein level of TLR4 and RelA (as an activator of NF-kB) need to be included here.
5) Some key references need to be added to boost the discussion part. For instance PMID: 33718383, PMID: 33969874.
Round 2
Reviewer 2 Report
The authors have partly addressed my previous experimental concerns. However, in view of the robustness associated with the study, I do accept their explanations.